# TREE2TREE LEARNING WITH MEMORY UNIT

## ABSTRACT

Traditional recurrent neural network (RNN) or convolutional neural network (CNN) based sequence-to-sequence model can not handle tree structural data well. To alleviate this problem, in this paper, we propose a tree-to-tree model with specially designed encoder unit and decoder unit, which recursively encodes tree inputs into highly folded tree embeddings and decodes the embeddings into tree outputs. Our model could represent the complex information of a tree while also restore a tree from embeddings.

We evaluate our model in random tree recovery task and neural machine translation task. Experiments show that our model outperforms the baseline model.

## 1 INTRODUCTION

In the past several years, Sequence-to-Sequence (Seq2Seq) model based on neural networks has made significant progress. It is widely used in various tasks, such as time series analysis(Lipton et al., 2015), speech recognition(Graves et al., 2013) and machine translation (Bahdanau et al., 2014). However, such model only captures sequential information while structural information is also of great importance in many tasks. On the one hand, data of many tasks is in hierarchical structure, such as interpersonal relationships in social network. Sequential data may contain latent structural information, on the other hand. For instance, natural language can be parsed into a tree with hierarchical structure, which represents the semantic dependency relationship between words (Chen & Manning, 2014). Dyer et al. (2016) has shown that introducing structural information into sentence embedding could lead to an improvement to neural machine translation.

General Seq2Seq model can be regarded as a transduction from sequential inputs to sequential outputs, which limits the data structure of applications. But there are a lot of data hierarchical structure that can be represented in tree instead of sequence format. Compared with sequence based model, tree based model is more difficult to construct for the following two reasons: first, the number of child nodes is usually uncertain; second, the structural information of a tree is difficult to represent and restore. Thus, constructing a more informative embedding in tree based model remains a challenging problem. Lots of efforts have been made to tackle this problem. Socher et al. (2011b) designed a recursive tree encoder to encode both content and structure information. Tai et al. (2015) introduced a LSTM tree encoder to calculate an embedding for classification tasks. However, few works are devoted to recovering structural information from a embedding of tree. Socher et al. (2011a) proposed a tree-based model which applied hierarchical perceptron in both encoding and decoding process as a recursive auto-encoder. But its cell cannot handle the long-term dependency.

To address this issue, we propose a novel tree-to-tree model that recursively encodes tree inputs into highly folded tree embeddings and decodes the embeddings into tree outputs. Our model could represent the complex information of a tree while also restore a tree from embeddings. Furthermore, inspired by the architecture of long short-term memory (LSTM) (Hochreiter & Schmidhuber, 1997), we proposed two memory units for encoder and decoder, respectively. Units of encoder contain an output gate, an input gate and forget gates, and units of decoder have an output gate as well as memory gates. Concretely, during encoding process, information from memory units of child nodes is combined in an organic way with the label of parent node, which is assigned to memory units of parent nodes. After getting the embedding, the decoder transduce it into a tree by splitting information into several parts and passing them to generate child nodes.

Besides, the similarity of two trees is hard to define. To alleviate this issue, we take both content and structure similarity into consideration and design a novel objective function. The content loss

reflects whether the corresponding nodes of two trees share the same label, while the structure loss measures the difference of child number for corresponding nodes. The final objective function is the weighted sum of content loss and structure loss.

We design a random tree recovery experiment to evaluate the performance of the tree-to-tree model to represent and restore tree structure data. Experiment result in Section 4 shows, our model can recover nearly 90% of node labels and nearly all structural information, which significantly outperform the previous work. To evaluate our model in real-world problems, we perform experiments in neural machine translation task. Our model shows an improvement on both performance and robustness.

Our main contributions are three-fold:

- We propose a tree-to-tree model with specially designed encoder unit and decoder unit.
- We design a novel objective function with content loss and structure loss for tree.
- Experiments on both tree recovery evaluation and machine translation shows a great improvement.

The complete training and testing code are available at http://www.panda.com

## 2 RELATED WORK

Tree-based models are widely used in recent AI research thanks to its ability to restore structure information which has been proved to help raise up performance in practice, like Wang et al. (2011) in image retrieval and Mou et al. (2015) in Heuristic Matching.

Recursive Neural Network(RNN) models are a popular family of tree-based deep learning models especially for NLP research. The main difference among these models lies in how the tree structure is composited, i.e., how to obtain the representation for a phrase or sentence with the representations of words it consists of. The Socher et al. (2011b) propose a basic form of RNN models, which represent phrases and words as vectors. It employ matrices and tensors as operators to perform compositions in a bottom-to-up way and finally reach a sentence embedding vector. Recently, there have been several studies addressing more semantic composition for RNNs. RNTN (Socher et al., 2013b) employs a tensor to model quadratic form interactions rather than basic matrix multiplication. MV-RNN (Socher et al., 2012) assigns each word with a vector and a matrix specifically. Also there have been studies on untied parameter tree composition including Socher et al. (2013a) and Hermann & Blunsom (2013) which utilize different composition functions according to the combined nodes' syntactic tags according to parsing results. Recently an adaptive framework were proposed (Dong et al., 2016) which selects composition function based on only the input vectors' value of their chid nodes.

As a combination of recursive neural network and LSTM, Tree-LSTM was first proposed in Zhu et al. (2015). It utilizes a memory cell to reflect the history memories of multiple child cells or multiple descendant cells recursively. Following its step, it is applied in NLP tasks and outperform the state-of-art result including language inference in Chen et al. (2017; 2016), sentiment classification and paraphrase analysis (Tai et al., 2015). In the above models, the Tree-LSTM structure is applied in the sentence encoding process. For decoding and tree generation process (Zhang et al., 2015) applies top-down Tree-LSTM to generate dependency parse tree.

As the aspect of tree language model, Brychcin (2016) proposes a model that construct a tree structure for different sentences without utilizing syntax information like parsing trees. Bechet et al. (2001) and Richardson et al. (2016) derive a tree language model dependent on dependent parsing trees. As far as we know, we are the first to take consistency parsing information into consideration when constructing a tree-based language model.

## 3 MODEL

In this paper, we introduce a novel tree-to-tree architecture, which first encodes the source tree as a context vector by a carefully designed encoder and to decode the vector to a target tree by a decoder. The overall workflow is shown in Table 1.

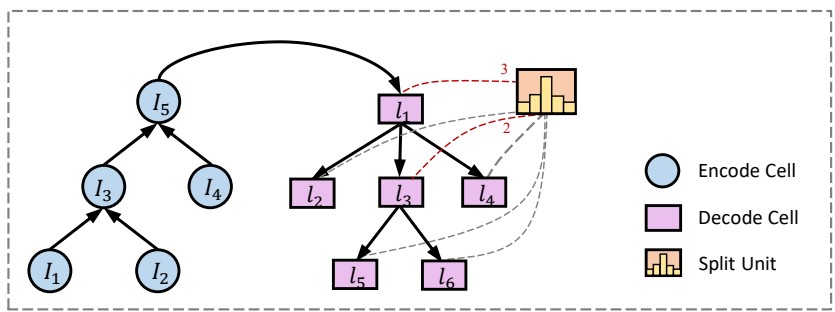

Figure 1: Tree2Tree Structure. Encoder cell get information from input and child nodes. Decoder uses state of root node of encoder($I_5$) as the state of root node of decoder($l_1$). For nodes in decoder, we first decide the number of child cells by split units. For example $l_1$ and $l_3$ have 3 and 2 child nodes, while other nodes have none. Then nodes split information for their children.

### 3.1 TREE MEMORY ENCODER(TME)

As intuitively showed in Figure 2(a), our encoder is improved from Socher et al. (2011a), which ensures that important information from nodes of lower level can be transmitted to the root node without much decay. In order to reconstruct trees from representation vectors generated by encoder, we designed a novel tree LSTM decoder. At each node of the decoder, information is properly devided into several parts and passed to its child nodes. The structure of Tree2Tree LSTM is shown in Figure 1.

While Socher et al. (2011a) utilize LSTM encoder to encode trees into a representation vectors containing corresponding node and structure information, we employ a recursive decoder to transform the representation vectors back to trees.

We define tree-encoder as a collection of recursively organized encoder cells. Given the tree structure and the inputs of the leaf nodes, denoted as $x_1, x_2, \cdots, x_n$ each represents an N-dimension vector. Each time a parent vector $p$ is computed from its child nodes $c_1, c_2, \cdots$ with equation .

$$p = TME([c_1; c_2; \cdots])$$

Here $[c_1; c_2; \cdots]$ denotes the concatenation of the children. From bottom to up, all the nodes are computed recursively.

Concretely, our TME unit is shown as follows.

$$
\begin{aligned}
i_t &= \sigma(\sum_{k=1}^{n} W_{(k)}^{(i)} h_{t-1}(k) + u^{(i)} I_t + b^{(i)}) \\
f_t^{(p)} &= \sigma(\sum_{k=1}^{n} W_{(k)}^{(f)(p)} h_{t-1}(k) + u^{(f)(p)} I_t + b^{(f)(p)}) \\
c_t &= \sum_{k=1}^{n} f_t^{(k)} \odot c_{t-1}(k) + i_t \odot I_t \\
o_t &= \sigma(W^{(o)} c_t + b^{(o)}) \\
h_t &= c_t \odot o_t
\end{aligned}
\tag{1}
$$

$c_t(i), h_t(i), (i = 1, 2, \cdots, n)$ are the memory and output vectors of the child units. $I_t$ the input. $f_t(i), (i = 1, 2, \cdots, n)$ are forget gates determining how much information from $c_t(i)$ will be discarded. $i_t$ is the input gate deciding which part of $I(i)$ can get into $c_t$. Since the value of gates vary for different cell, information can be maintained for a long time.

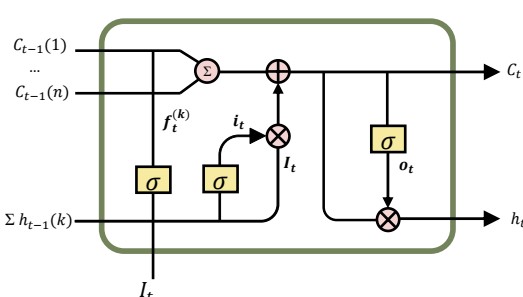
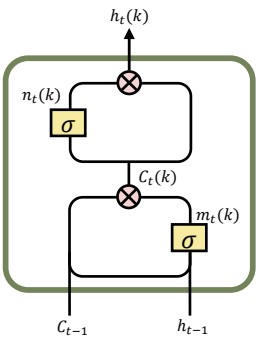

(a) The Structure of Encoder cell. $C_{t-1}(k)$ is values of child node's memory units. $I_t$ is the input. After passing through forget gate $f_t^{(k)}$ and input gate $i_t$, information from $C_{t-1}(k)$ and $I_t$ are combined to generate new cell state and hidden state, $c_t$ and $h_t$.

(b) Generation process for one of the child units. $C_{t-1}$ and $h_{t-1}$ are cell state and hidden state of parent cell. After passing through memory gate $m_t(k)$ for the k-th child node, a specific part of information from $C_{t-1}$ is transmitted to the k-th child node.

Figure 2: Units of Our Model

## 3.2 TREE MEMORY DECODER(TMD)

While tree-encoder is natural extention of sequence LSTM cell, there is a large difference between tree-decoder and sequence LSTM decoder. Sequence LSTM encoder only only need to recover content information, but tree-decoder should also resolve structure information. That is to say, a tree-decoder cell need to decide what to output, as well as the number child cells. So the structure of tree-decoder is much much complicated that sequence-LSTM cell. The structure of decoder is shown in Figure 2(b).

$$
\begin{aligned}
nu\_prob_t &= softmax(W_c^{(nu)}c_t + W_h^{(nu)}h_t + b^{(nu)}) \\
nu_t &= \arg\max(nu\_prob_t) \\
m_t(k) &= \sigma(W_c^{(m)(k)}c_t + W_n^{(m)(k)}h_t + b^{(m)(k)}) \\
c_t(k) &= m_t(p) \odot c_t \\
n_t(k) &= \sigma(W^{(o)}c_t(k) + b^{(o)}) \\
h_t(k) &= c_t(k) \odot o_t(k)
\end{aligned}
\tag{2}
$$

where $nu_t$ is a gate decide how much child nodes to generate.

$m_t(i)(i = 1, \cdots, nu_t)$ are memery-keep cells determining how information is splited from $c_t, n_t(i)(i = 1, \cdots, nu_t)$ are output gates of child nodes, and $h_t(i), (i = 1, \cdots, nu_t)$ are the corresponding outputs. At each step cells decide what to output, how many children to generate and the proportion of remaining information to pass to a specific child node. According to the spilt number $nu_t$ determined by our decoding unit at each time, we generate the child nodes $c_1, c_2, \cdots, c_{nu_t}$ through the parent node with equation .

$$
[c_1; c_2; \cdots; c_{nu_t}] = TMD(p)
\tag{3}
$$

## 3.3 OBJECT FUNCTION

There are two criterion to assess the ability of autoencoder - the correctness of structure and precision of content recovery. So we need to minimize structure loss $L_1$ and content loss $L_2$ at the same time. We define $L_1$, $L_2$ as the cross entropy of generating the right number of child nodes and output the

right token. We use $L = L_1 + \lambda L_2$ as our training loss. And we use Adam(Kingma & Ba, 2014) as the optimizer, which is more efficient and convenient to use than SGD.

For the target tree $T_r$ and the output tree $T_o$,

$$L_{content} = -\sum_{i \in T_r} \left( I(i, T_r) \cdot \left( \sum_{j \in T_o} \left( h(i,j) \log(label_{T_r(i)} \cdot label_{(T_o(j))}) \right) \right) \right) \quad (4)$$

$$L_{structure} = -\sum_{i \in T_r} \left( I(i, T_r) \cdot \left( \sum_{j \in T_o} \left( h(i,j) \log(childnum_{T_r(i)} \cdot childnum_{T_o(j)}) \right) \right) \right) \quad (5)$$

where,

$$I(i, T) = \begin{cases} 1, \text{if} \exists j \in T, \text{the position of } i \text{ is same as } j, \\ 0, \text{otherwise} \end{cases}$$

$$H(i, j) = \begin{cases} 1, \text{if the position of } i, j \text{ are same}, \\ 0, \text{otherwise} \end{cases}$$

$label_{T_r(i)}, childnum$ means the label and child number of node $i$ in tree $T_r$. Node $i$ and node $j$ have the same position if and only if the position of the parent and left brother of $i, j$ are same.

## 4 EXPERIMENT

To demonstrate the effectiveness of our model, we test our model on two tasks, which includes the tree reconstruction task and machine translation task.

### 4.1 EXPERIMENT SETUP

**Dataset**

For for tree reconstruction task, we choose the mathematics genealogy dataset [1]. Each tree represents a teacher-student relationship of mathematicians. A node indicate a person with its child nodes as the person's direct students. Each node is represented by an integer label. For nodes with more than 2 child nodes, we split the child nodes with each cluster sized 2, and construct corresponding trees. Finally we got 10000 binary trees. We split all these data into 3 parts, with 9000, 500, 500 each for training/validation/test.

For the machine translation task, our data set consists of 350,000 sentences from United Nations Parallel Corpus (Ziemski et al., 2016). We choose the first 50,000, 100,000, 340,000 sentence pairs as training set at each time, the validation set and test set both consist of 5,000 sentences. For Chinese corpus, Jieba [2] is applied in tokenization. We use Stanford parser (Klein & Manning, 2003; Levy & Manning, 2003) to get parsing tree of each sentence.

**Implementation Details**

All our experiments are run on GPUs in tensorflow framework. The autoencoder is trained on our model and the baseline separately with the training set. All parameters in our model are initialized with a uniform distribution within $[-0.01, 0.01]$.

For machine translation tasks, our dictionary consists of all the words occurring in the training set, with each word vector initialized via Glove (Pennington et al., 2014) method in 300 dimensions and set to be joint trained along the model weights.

### 4.2 EXPERIMENT RESULTS

**Tree Reconstruction**

In this part, we design a self-to-self task to evaluate our model's recovery ability on arbitrary trees. Our model is compared with recursive Auto-Encoder (RAE) (Socher et al., 2011a).

---

[1] https://genealogy.math.ndsu.nodak.edu/
[2] https://pypi.python.org/pypi/jieba/

We evaluate the performance of recovery on label accuracy and structure accuracy. The label accuracy reflects how many nodes are correctly predicted, and the structure accuracy denotes the percentage of trees whose predicted structure is the same as the reference. Our Tree2Tree model reaches 85.9% in labeled accuracy and 100% in structure annuracy, far higher than the baseline which is 73.4% and 79.6%. The exciting result reflects that our tree2tree perfectly maintain almost all of the structured information when it is utilized as an autoencoder.

Figure 4.2 shows a example of tree reconstruction result. we can observe that our model reconstructs the original tree much better than the baseline in both node level and structure level.

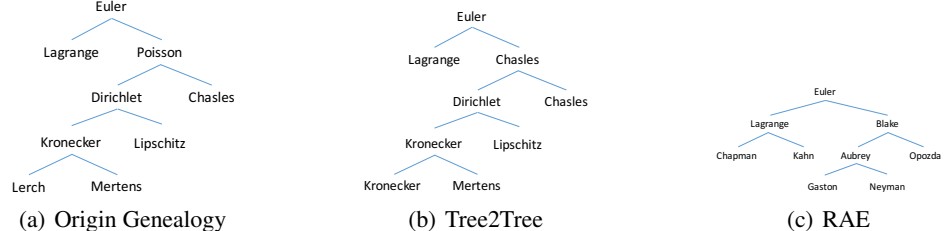

(a) Origin Genealogy      (b) Tree2Tree      (c) RAE

Figure 3: A example of tree reconstruction result. (a) is the reference tree. (b) and (c) denote the reconstruction result of our Tree2Tree model and the baseline model (Socher et al., 2011a), respectively.

**Machine Translation**

In this part, we compare our model against the lstm baseline on English to Chinese machine translation task.

Table 1 shows Bleu (Papineni et al., 2002) scores in test set of our model and LSTM of English to Chinese translation task. Here the loss function in our model is defined as the cross entropy between prediction and target. We report results of both our model and LSTM with the same random seed and pick the one with best validation perplexity for final Bleu evaluation. Translation are generated via beam search with beam size 5. We can see that our tree2tree model outperforms the baseline for each data scale, with obvious difference in small dataset.

|  | 5w | 10w | 35w |
|---|---|---|---|
| LSTM | 0 | 15.9 | 28.9 |
| Tree2Tree LSTM | 13.4 | 18.7 | 29.3 |

Table 1: Reslut of Translation

|  | BLEU1 Reduction(%) | BLEU2 Reduction(%) |
|---|---|---|
| LSTM | 23.3 | 32.5 |
| Tree2Tree LSTM | 18.9 | 30.9 |

Table 2: Robustness Analysis

We also test the robustness of our model against the baseline. For each sentence, we replace one word randomly in the original text with a word randomly chosen in the dictionary, and evaluate the difference of the output to test the robustness. We use the reduction proportion of Bleu1 and Bleu2 scores to evaluate the robustness of translation. The result is shown in Table 2, from which we can see that Tree2Tree LSTM are more stable than LSTM. We will prove this property in theory in Section 5.

# 5 ANALYSIS

In this section, we intruduce a Neural Tree Generation Model as a theoretical analysis which provides an explanation for our Tree2Tree in view of probability.

For tree $T$, we assume that the generation probability of node $w_i \in T$ satisfies the conditional independent property as follows.

$$\forall \text{node} \quad w_j \neq w_i, w_j \perp w_i | par(w_i)$$

Then, we can get the generation probability of the tree in the Figure 4(c).

$$P(T) = P(r_3|r_1)P(r_4|r_1)P(r_1|r_0)P(r_2|r_0)P(r_0)$$

Table 3: Examples of Robustness

| Original Text |
| --- |
| consideration of reports submitted by states parties under article 16 of the convention |
| financing of the united nations preventive deployment force |
| second periodic reports due in 1996 |
| these figures show that womens participation in politics continues to be limited |
| the group of experts had before it the following documents |

| Reference |
| --- |
| 审议缔约国在盟约第16 条下提交的报告 |
| 联合国预防性部署部队经费的筹措 |
| 应于1996 年提交的第二次报告 |
| 这些数字表明妇女参与政治生活继续受到限制 |
| 专家组收到下列文件 |

| LSTM |
| --- |
| 审议缔约国提交的报告提交的国家间工作组的报告 |
| 联合国预防性部署部队联预部队 |
| 1996 年11 月10 日 |
| 这些数字表明妇女继续在这些部门中占主导地位 |
| 专家组的以下文件是的的的工作 |

| Tree2Tree LSTM |
| --- |
| 审议缔约国在公约第16 条下提交的报告 |
| 联合国预防性部署部队经费的筹措 |
| 应于1996 年第二次定期定期报告 |
| 这些数字表明认为参与妇女人数仍然有限 |
| 专家组有以下文件 |

Formally, we can define the generate probability of a tree by equation 6.

$$P(tree) = \prod_{w_i \, \text{node}:w_i} P(w_i|par(w_i)) \tag{6}$$

Considering the tree structure as a Bayesian Network and every node as a random vector, we can get the set of Independence assertions of the tree.

$$I(tree) = \{w_i \perp NonDescendant_{w_i}|par(w_i)\} \tag{7}$$

Where $NonDescendant_{w_i}$ denotes the nodes in the tree that are not descendants of $w_i$.

According to equation 7, we can find that for two nodes with different parent nodes, the probability of two nodes is independent when parent nodes of two nodes are given. Formally, the tree structure can be regarded as a Bayesian Network(Koller & Friedman, 2009), which leads to the independencies in the tree-structure using D-separation. For instance, Figure 4(c) provides an intuitive example that $(r_3 \perp r_2|r_1)$.

In NLP tasks, the model above can be regarded as a neural tree-structure language model. For a sentence $(w_1, w_2, \cdots, w_n)$, we define the probability of $P(w_1, w_2, \cdots, w_n)$ as follows.

$$P(w_1, w_2, \cdots, w_n) = P(\text{parsing tree of } (w_1, w_2, \cdots, w_n)) \tag{8}$$

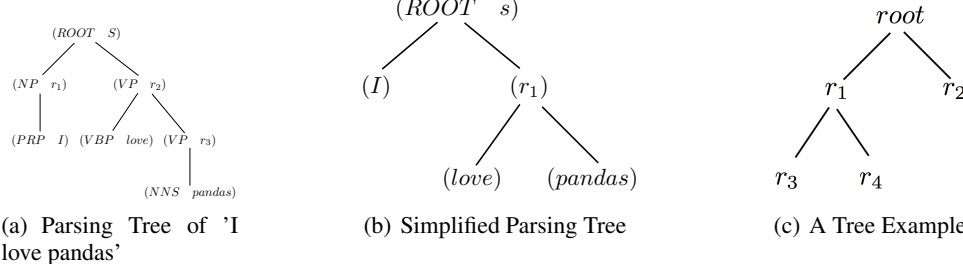

(a) Parsing Tree of 'I love pandas'

(b) Simplified Parsing Tree

(c) A Tree Example

Figure 4: Parsing Tree of Example Sentence

where $par(w_i)$ denotes the parent node of $w_i$ in the parsing tree of the sentence.

Take 'I love pandas' for instance, the parsing tree of the sentence is shown in Figure 4, the probability of the sentence is computed as follows(by equation 9).

$$P('I','love','pandas') = P('I'|r_1)P('love'|r_2)P('pandas'|r_2)P(r_1|r_0)P(r_2|r_0)P(r_0) \quad (9)$$

The parsing tree of a sentence is constructed by the grammar of the sentence, which shows the syntactic structure of the sentence. This is consistent with the independency we get from Bayesian Network.

Besides, neural tree-structure language model have an another advantage. LSTM regards a sentence with a length of $n$ as a sequence of $n$ words. We define the path length of a word as the number of units the word goes through before reaching the end node of sentence embedding. For LSTM, the end node is the last word of sentence, and for tree-structured model, the end node is the root node of a tree. The AVE(average path length) of LSTM is $O(n)$, but the AVE of tree LSTM is $O(\log(n))$. So tree LSTM usually has smaller AVE.

## 6 CONCLUSION AND FUTURE WORK

In this paper, we introduce a novel Tree2Tree LSTM which contains a tree-structure encoder and a tree-structure decoder. The model can apply to many fancy tasks, such as machine translation, tree-structure generation, etc. Based on our model, some experiments and theroy analysis are done to prove the strength of our model. In the future work, we will try to append tree-based attention to our model to improve the performance of our model.

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
