# OpenReview forum: "Tree2Tree Learning with Memory Unit"
_ICLR.cc/2018/Conference — Reject_

### Official Review · AnonReviewer2 · 2017-11-27

**Rating:** 2
**Confidence:** 4

**Review:**

Summary: the paper proposes a tree2tree architecture for NLP tasks. Both the encoder and decoder of this architecture make use of memory cells: the encoder looks like a tree-lstm to encode a tree bottom-up, the decoder generates a tree top-down by predicting the number of children first. The objective function is a linear mixture of the cost of generating the tree structure and the target sentence. The proposed architecture outperforms recursive autoencoder on a self-to-self predicting trees, and outperforms an lstm seq2seq on En-Cn translation.

Comment:

- The idea of tree2tree has been around recently but it is difficult to make it work. I thus appreciate the authors’ effort. However, I wish the authors would have done it more properly.
- The computation of the encoder and decoder is not novel. I was wondering how the encoder differs from tree-lstm. The decoder predicts the number of children first, but the authors don’t explain why they do that, nor compare this to existing tree generators.
- I don’t understand the objective function (eq 4 and 5). Both Ls are not cross-entropy because label and childnum are not probabilities. I also don’t see why using Adam is more convenient than using SGD.
- I think eq 9 is incorrect, because the decoder is not Markovian. To see this we can look at recurrent neural networks for language modeling: generating the current word is conditioning on the whole history (not only the previous word).
- I expect the authors would explain more about how difficult the tasks are (eg. some statistics about the datasets), how to choose values for lambda, what the contribution of the new objective is.

About writing:
- the paper has so many problems with wording, e.g. articles, plurality.
- many terms are incorrect, e.g. “dependent parsing tree” (should be “dependency tree”), “consistency parsing” (should be “constituency parsing”)
- In 3.1, Socher et al. do not use lstm
- I suggest the authors to do some more literature review on tree generation

---

### Official Review · AnonReviewer3 · 2017-11-27
**The paper is not ready for publication yet. It has very limited contributions and evaluation is preliminary.**

**Rating:** 5
**Confidence:** 4

**Review:**

This paper proposes a tree-to-tree model aiming to encode an input tree into embedding and then decode that back to a tree. The contributions of the work are very limited.  Basic attention models, which have been shown to help model structures, are not included (or compared). Method-wise, the encoder is not novel and decoder is rather straightforward. The contributions of the work are in general very limited. Moreover, this manuscript contains many grammatical errors.  In general, it is not ready for publication.

Pros:
- Investigating the ability of distributed representation in encoding input structured is in general interesting. Although there have been much previous work, this paper is along this line.

Cons:
- The contributions of the work are very limited. For example, attention, which have been widely used and been shown to help capture structures in many tasks, are not included and compared in this paper.
- Evaluation is not very convincing. The baseline performance in MT is too low. It is unclear if the proposed model is still helpful when other components are considered (e.g., attention).
- For the objective function defined in the paper, it may be hard to balance the "structure loss" and "content loss" in different problems, and moreover, the loss function may not be even useful in real tasks (e.g, in MT), which often have their own objectives (as discussed in this paper). Earlier work on tree kernels (in terms of defining tree distances) may be related to this work.
- The manuscript is full of grammatical errors, and the following are some of them:
"encoder only only need to"
"For for tree reconstruction task"
"The Socher et al. (2011b) propose a basic form"
"experiments and theroy analysis are done"

---

### Official Review · AnonReviewer1 · 2017-11-27
**Please compare with other methods**

**Rating:** 4
**Confidence:** 4

**Review:**

This paper presents a model to encode and decode trees in distributed representations.
This is not the first attempt of doing these encoders and decoders. However, there is not a comparative evalution with these methods.
In fact, it has been demonstrated that it is possible to encode and decode trees in distributed structures without learning parameters, see "Decoding Distributed Tree Structures" and "Distributed tree kernels".
The paper should present a comparison with such kinds of models.

---

### Public Comment · ~bruce_matthew_kuzak1 · 2017-11-29
**Class Project**

We are tasked to evaluate a research paper as a class project and we need to evaluate your results and their plausibility. Could we have access to your source code for training the Neural Nets and the training data to analyze the results.

It would be of great help and we could forward a lot of positive feedback hopefully.

---

### Decision · Program_Chairs · 2018-01-29
**ICLR 2018 Conference Acceptance Decision**

**Decision:**

Reject

**Comment:**

the problem is interesting, and the reviewers acknowledge it's worth an effort to tackle. unfortunately all the reviewers found the work to be too preliminary without a convincing evidence supporting the proposed approach against other alternatives (or on its own.)